# The Effects of Contrast between Dark- and Light-Coloured Tanks on the Growth Performance and Antioxidant Parameters of Juvenile European Perch (*Perca fluviatilis*)

Áron Molnár [1,2], Dávid Zoltán Homoki [2], Péter Bársony [3,*], Attila Kertész [1,2], Judit Remenyik [4], Georgina Pesti-Asbóth [4] and Milán Fehér [1]

1   Department of Animal Husbandry, Institute of Animal Science, Biotechnology and Nature Conservation, Faculty of Agricultural and Food Sciences and Environmental Management, University of Debrecen, 4032 Debrecen, Hungary; molnar.aron@agr.unideb.hu (Á.M.); kertesz@agr.unideb.hu (A.K.); feherm@agr.unideb.hu (M.F.)

2   Doctoral School of Animal Science, University of Debrecen, 4032 Debrecen, Hungary; homokidz@agr.unideb.hu

3   Department of Animal Nutrition Psychology, Institute of Animal Science, Biotechnology and Nature Conservation, Faculty of Agricultural and Food Sciences and Environmental Management, University of Debrecen, 4032 Debrecen, Hungary

4   Institute of Food Technology, Faculty of Agricultural and Food Sciences and Environmental Management, University of Debrecen, 4032 Debrecen, Hungary; remenyik@agr.unideb.hu (J.R.); georgina.asboth@agr.unideb.hu (G.P.-A.)

*   Correspondence: barsonp@agr.unideb.hu

**Abstract:** European perch (*Perca fluviatilis*) is a predatory fish species with a high degree of stress sensitivity during rearing in intensive systems. Our study was focused on the effects of contrast between two colours (black and light grey) in different parts of a tank (bottom and sidewall) on the production and antioxidant parameters of juvenile European perch during intensive rearing. The duration of the experiment was 8 weeks. In the first treatment, the bottoms of the tanks were black (DB); in the second treatment, the sides of the tank were black (DS); in the control treatment, the fish were kept in light-grey tanks (K). There were three replicates per treatment, and a total of 180 individuals were used; therefore, 60 individuals were used per treatment, with 20 individuals per tank. The mean body weight of the fish at the start of the experiment was 32.01 ± 0.79 g. At the end of the experiment, the antioxidant parameters (cortisol, glucose, MDA, catalase, vitamin C, GPx, GR, GSH, GSSG, and HSP70) of the fish were determined from blood samples. The results of our experiment show that different levels of contrast between the dark and light tank colours significantly influenced the production and antioxidant parameters of the juvenile European perch.

**Keywords:** RAS; European perch; tank colour; contrast; black; light grey; reactive oxygen species; antioxidant parameters; stress level

## 1. Introduction

European perch (*Perca fluviatilis*) plays an increasingly important role in aquaculture production today [1], and the European demand for this species is growing [2]. At present, the main aquaculture producers in Europe are currently Switzerland, Russia, France, Poland, Ireland, Belgium, and Denmark. In 2019, the global capture of European perch reached 32,000 tonnes; in line with this, the annual global European perch aquaculture production has increased, but it only was 954 tonnes [3]. This trend is supported by the fact that aquaculture European perch production has more than doubled in the last 5 years [3] owing to RAS systems. In order to meet the growing demand, it is necessary to increase the volume of aquaculture European perch production, which can be achieved only in a recirculation aquaculture system (RAS). The recirculation system allows

precision fish production under controlled conditions with minimum water consumption. Under intensive conditions, optimisation of farming technology is required to increase production efficiency.

European perch is a predatory fish species, but its diet also includes small invertebrates [4,5]. It is characterised by the disadvantage of slow growth but has the desirable attribute of excellent meat quality [6,7]. However, the market size (120–140 g) of the fish can be achieved in one year by ensuring optimal environmental conditions (water temperature and oxygen level, water quality, light conditions,) under intensive conditions [6]. The optimal temperature, light intensity, and oxygen saturation are 22–24 °C, 200–1.100 lx, and 60–72%, respectively. The ammonia and nitrite level has to be below 0.3mg $N-NH_3$ $L^{-1}$ and 0.5 mg $N-NO_2$ $L^{-1}$ [8].

European perch is a sensitive fish species that responds to various environmental factors with stress responses that affect production parameters in all kinds of rearing systems [9,10]. In order to optimise the efficiency of rearing, minimisation of the effects of stress on the fish is of particular importance. Therefore, during rearing, suitable conditions have to be provided, for example, by reducing disturbance [11]. Under controlled conditions, stress effects can be reduced by changes in farming techniques, such as modifying the colour of the tank or part of it [12,13].

For several fish species, it has been shown that the tank colour fundamentally influences the stress status of fish and thus their production parameters [14–16]. Strand et al. [17] found that a completely black tank is not optimal for rearing juvenile European perch. In their experiment, white, grey, and black tanks were compared. It was found that fish kept in white and grey tanks showed better production parameters than those kept in black tanks. In accordance with their results, Tamaouzt et al. [14] concluded that European perch larvae showed better growth performance in tanks with light grey and white walls than in tanks with black walls. Contrary to these results, during larvae rearing of European perch, the dark tank colour shows better production parameters, survival rate, and swim bladder inflation [13,18]. Based on these results, it can be stated that different tank colours are recommended for the rearing of European for different development stages.

The feeding behaviour of European perch and the environmental light conditions were found to be strongly connected [19]. Different light intensities also play major roles in the effects of stress in European perch [20–22]. The colour of the tank fundamentally determines the light intensity inside the growing unit. In this research, different tank colours were used, since our study focused on the contrast between two colours (black and light grey) in different parts of the tank (the bottom and sidewalls).

The high stress sensitivity of European perch is well known [23], which negatively affects production parameters (body weight, specific growth rate, survival rate, feed conversion ratio). Reactive oxygen species (ROS) are excellent biomarkers for measuring the stress status of fish. Many factors influence the formation of ROS in fish. The stress responses of fish species that are triggered by particular situations may be caused by several environmental biotic and abiotic factors [24–31]. Although a number of researchers have examined ROS, none has investigated the relationship between reactive oxygen species (ROS) and tank colours in the case of fish. Oxidative stress is the result of an imbalance between free radicals and antioxidants [32]. Oxidative stress can play a key role in the complex coping mechanisms used by fish to adapt to changing environmental impacts [33].

Organisms, in general, do not aim to totally eliminate ROS but, rather, to optimise the balance between pro- and antioxidants. When the balance shifts towards pro-oxidants, there is a situation of oxidative stress. To manage oxidative stress, organisms establish a three-level antioxidant defence system [34]. The first level of defence is antioxidant enzymes (catalase and glutathione peroxidase), and the second level is low-molecular-weight antioxidants (glutathione, reduced glutathione, oxidised glutathione, malondialdehyde, and vitamin C) [32]. The third and final level is activated only when damage has already occurred, and its role is to remove or repair damaged parts, such as heat shock proteins

(HSP70) or glutathione reductase. The role of HSP70 is to maintain cellular homeostasis in the case of sudden environmental changes [35–37].

In addition to the above-mentioned antioxidant parameters, the cortisol and glucose levels of the fish, which are well-known, commonly used biomarkers for measuring stress status in fish [38–44]. Cortisol is a hormone that is frequently examined as a stress indicator. Plasma cortisol levels increase in response to stressors [45]. Cortisol production during a stress response promotes an increase in glucose concentration [38].

Many scientific papers are available about recommended tank colours for European perch farming at different developmental stages, but the results are often contradictory. Therefore, the aim of our experiment was to determine the optimal combinations of tank colours for juvenile European perch rearing.

In our study, the effects of a light-grey tank, a tank with a black bottom and light-grey sidewalls, as well as a tank with a light-grey bottom and black sidewall on production (body weight, specific growth rate, survival rate, feed conversion ratio) and antioxidant parameters (cortisol, glucose, MDA, catalase, vitamin C, GPx, GR, GSH, GSSG, and HSP70) of juvenile European perch were investigated. In order to determine the stress conditions of the fish, the antioxidant parameters were measured according to each level of the three-level antioxidant system.

## 2. Materials and Methods

### 2.1. Experimental Design

This experiment was carried out at the Laboratory of Fish Biology of the University of Debrecen, Faculty of Agricultural, Food Science, and Environmental Management, and approved by the ethical approvement of DEMAB/15/2019. The duration of the experiment was 8 weeks.

The experimental recirculation system consisted of three main parts: the rearing units, as well as the mechanical and biological filters. Mechanical filtration was achieved using sponges, while biological filtration was executed by bio media followed by UV sterilisation.

In our experiment, the base colour of the tanks was light grey (control group, K). In addition to the control group, two other groups were applied with the combinations of the light-grey and black colours. In the first group, only the sidewalls of the tanks were black (DS), and in the second group, only the bottoms of the tanks were black (DB).

The initial mean body weight of the experimental stock was $32.01 \pm 0.79$ g. The experimental stock was derived from artificial propagation of H&H Carpio Fish Farming Ltd., Hungary, and kept in a recirculation system prior to the experiment. The experimental system consisted of 9 units, with a total volume of 350 L water and a total of 180 fish (3 tanks/treatment, 20 fish/tank, 60 fish/treatment). Each tank was equipped with an aeration stone to ensure the optimal oxygen level within the tank.

The fish were exposed to natural lighting conditions. Prior to beginning the experiment, the fish were acclimatised to the living conditions for 2 days. The daily feed portion was 1% of the total fish biomass (2 mm feed, 54% protein, and 20% fat, Aller Aqua Group, Allervej, Christiansfeld, Denmark) and was given twice a day by hand. Uneaten feed and fish faeces were removed daily from the bottoms of the tanks.

### 2.2. Water Quality Parameters

The dissolved oxygen content of the water ($7.55 \pm 0.34$ mg/L), the water temperature ($22.65 \pm 0.91$ °C), and pH ($7.51 \pm 0.14$) were checked daily (HACH HQ30d). The concentrations of the different forms of nitrogen in the water were measured weekly ($N\text{-}NH_3^+$ ($0.8 \pm 0.30$ mg/L), $N\text{-}NO_2^-$ ($0.07 \pm 0.11$ mg/L), and $N\text{-}NO_3^-$ ($12.6 \pm 6.1$ mg/L)) using a HACH DR3900 spectrophotometer (Hach Company CO, Ames, IA, USA).

### 2.3. Fish Growth Parameters

At the end of the experiment, the survival rate (S), final body weight ($BW_f$), specific growth rate (SGR), feed conversion ratio (FCR), Fulton condition factor (K), and homogeneity of the stock (CV%) were determined using the following formulas:

$$\text{Survival rate (\%) = (harvested individuals/stocked individuals)} \times 100 \tag{1}$$

$$\text{Specific growth rate (SGR) (\%/day) = (lnWf - lnWi)/t} \times 100 \tag{2}$$

where Wf is the wet final body weight (g), Wi is the wet initial body weight (g), and t is the time (day).

$$\text{Feed conversation ratio (FCR) (g/g) = F/(Wf - Wi)} \tag{3}$$

where F refers to the amount of dry feed consumed during the experiment (g), Wf refers to the final body weight (g), and Wi stands for the initial body weight (g).

$$\text{Fulton condition factor (K) = W/L3} \times 100 \tag{4}$$

where W is the wet weight (g), and L is the standard length (mm).

$$\text{Homogeneity (CV\%) = (standard deviation/mean)} \times 100 \tag{5}$$

### 2.4. Sample Collection

At the end of the experiment, three European perch per unit were euthanised for blood sampling using clove oil. The blood was taken from the caudal blood vessels in the fish with a 1 mL volume single-use insulin syringe. During sampling, 2 mL of pooled sample was collected into EDTA tubes, and a coagulation inhibitor (heparin) was added to prevent clotting. During the sample preparation, the EDTA tubes were centrifuged at 2500 rpm at 4 °C for 10 min in order to measure the antioxidant parameters. The plasma, which separated into the supernatant, was pipetted into Eppendorf tubes and stored, frozen, at −20 °C until further use.

### 2.5. ELISA Assay

The concentration of vitamin C (PRS-0060FI PARS BIOCHEM), cortisol levels (PRS-0006FI PARS BIOCHEM), and HSP70 levels (PRS-0026FI PARS BIOCHEM) were determined using commercially available ELISA kits. The glucose levels were determined using the commercially available E-Labscience kit (cat. no. E-BC-K234, Eching, Germany). Glutathione reductase (ab83461, Abcam), glutathione (Glutathione Assay Kit, item no. 703002), vitamin C (ab65656, Abcam), catalase (Catalase Activity Kit (ab83464, Abcam)), and glutathione peroxidase (ab102530, Abcam) were measured using a commercially available colorimetric kit. Malondialdehyde (MDA) concentration was determined using a commercially available assay kit (Abcam ab118970, Cambridge, UK). GR, GPx, GSH, GSSG, and MDA were quantified using a SPECTROstar Nano Microplate reader (BMG LABTECH, Ortenberg, Germany).

### 2.6. Statistical Analysis

Statistical analyses were performed using SPSS 22.0 for windows software. The obtained data were tested for normality of distribution by Kolmogorov–Smirnov test. The homogeneity of variances between experimental groups was checked by Levene's test. The effects of the treatments on the growth performance and antioxidant parameters of fish were analysed by one-way analysis of variance (ANOVA). Tukey's multiple comparison test (initial and final body weight) and Duncan's multiple range test (S, FCR, SGR, CV, K) were used to determine the significant differences between treatments [46–48]. $p < 0.05$ was considered significant for all analyses.

## 3. Results

### 3.1. Production Parameters

During the entire experiment, no mortality was observed; hence, the survival rate was 100% for all the setups.

The results obtained for the final body weight (BWf), the specific growth rate (SGR), and the feed conversion ratio (FCR) are presented in Table 1. The individual body weights of the fish (BWf) reared in the DB group were significantly higher than those of the fish reared using other treatments ($p < 0.05$). Moreover, there was no significant difference in BWf between the DS group and the control group. The SGR was significantly higher in the DB group than that in the DS group; however, the SGRs in the treatment groups did not differ significantly from those of the control group. Presumably, the dark bottom was similar to the natural habitat of European perch. Regarding the FCR values, the condition factor (K), and the homogeneity of the stocks (CV%), no differences were observed between the groups during the trial.

**Table 1.** Production parameters of European perch (*Perca fluviatilis*) kept in dark-sided (DS) and dark-bottomed (DB) tanks for 8 weeks.

| | Experimental Treatments | | |
|---|---|---|---|
| **Parameters** | **Control** | **DS** | **DB** |
| S (%) | 100 | 100 | 100 |
| BWf (g) | 49.22 ± 11.21 [a] | 48.70 ± 11.21 [a] | 54.24 ± 9.70 [b] |
| SGR (%/day) | 0.84 ± 0.14 [a,b] | 0.76 ± 0.08 [a] | 1.00 ± 0.06 [b] |
| K | 2.62 ± 0.25 | 2.66 ± 0.25 | 2.70 ± 0.27 |
| FCR (g/g) | 1.38 ± 0.24 | 1.40 ± 0.19 | 1.29 ± 0.13 |
| CV% | 22.40 ± 6.56 | 22.44 ± 4.66 | 17.51 ± 2.46 |

Data are presented as mean ± standard deviations. [a,b]: mean values within a row with different superscript letters are significantly different ($p < 0.05$).

### 3.2. Antioxidant Parameters

The effects of the different combinations of the two tank colours (DS, DB) on the antioxidant parameters of the fish are shown in Table 2.

**Table 2.** Antioxidant and stress parameters of European perch (*Perca fluviatilis*) kept in dark-sided (DS) and dark-bottomed (DB) tanks for 8 weeks.

| Antioxidant | Control | DS | DB |
|---|---|---|---|
| Cortisol (mg/mL) | 56.62 ± 2.89 [a] | 66.33 ± 2.08 [b] | 55.79 ± 3.20 [a] |
| Glucose (mmol/L) | 4.48 ± 0.95 [a,b] | 5.33 ± 0.94 [b] | 4.13 ± 0.83 [a] |
| Catalase (mU/mL) | 4.95 ± 0.51 [a] | 6.94 ± 0.51 [b] | 4.68 ± 0.91 [a] |
| Glutathione peroxidase (GPx) (mU/mL) | 182.52 ± 14.11 | 220.09 ± 37.62 | 169.48 ± 17.66 |
| Glutathione reductase (GR) (mU/mL) | 21.42 ± 3.94 [b] | 12.06 ± 1.13 [a] | 22.51 ± 5.02 [b] |
| Reduced glutathione (GSH) (µM) | 19.68 ± 2.09 [b] | 14.74 ± 3.93 [a] | 23.15 ± 0.57 [b] |
| Glutathione disulphide (GSSG) (µM) | 9.76 ± 1.07 [b] | 7.37 ± 1.96 [a] | 11.66 ± 0.26 [b] |
| Vitamin C (nmol/mL) | 46.54 ± 13.60 [a,b] | 30.38 ± 9.67 [a] | 92.32 ± 59.99 [b] |
| Malondialdehyde (MDA) (nmol/mL) | 1029.31 ± 183.83 [a] | 1604.59 ± 412.06 [b] | 1172.85 ± 239.08 [a,b] |
| HSP70 (ng/L) | 17.03 ± 0.85 | 17.11 ± 1.26 | 17.05 ± 0.94 |

Data are presented as mean ± standard deviations. [a,b]: mean values within a row with different superscript letters are significantly different ($p < 0.05$).

Cortisol and glucose, which are stress biomarkers, were measured. Fish reared in tanks with dark bottoms (DB) (55.79 ± 3.20 g) and the control group (56.62 ± 2.89 g) showed significantly lower cortisol concentrations than fish in the DS group (DS = 66.33 ± 2.08 g). The glucose levels were correlated with the cortisol levels in fish in the different groups. Fish in the DB group showed significantly lower glucose levels than fish in the DS group, while fish in the control group did not differ (DS = 5.33 ± 0.94 g; DB = 4.13 ± 0.83 g; control = 4.48 ± 0.95 g).

Among antioxidant enzymes, the catalase levels of the control group (4.95 ± 0.51 g) and the DB group (4.68 ± 0.91 g) were significantly lower than those of the DS group (6.94 ± 0.51 g); however, the GR values showed a completely opposite trend. The GR values in the DS group were significantly lower than those in the control and DB groups. No significant difference between the groups was found in the case of the third antioxidant enzyme (GPx) (DS = 220.09 ± 37.62 g, DB = 169.48 ± 17.66 g, C = 182.52 ± 14.11 g).

Regarding the different forms of glutathione (GSH, GSSG) that were measured, both the GSH and the GSSG levels did not differ between the control and DB groups, which were significantly higher than the levels in the DS group.

The results obtained for the low-molecular-weight antioxidants showed that the vitamin C levels of the fish were significantly higher in the DB group (92.32 ± 59.99 g) than those in the DS group (30.38 ± 9.67 g), while the levels in neither group significantly differed from those in the control group (46.54 ± 13.60 g). In the DB group, the highest vitamin C levels were measured in the blood samples, presumably due to the fact that the fish did not use up their bodily vitamin stores in defence against oxidative stress. Considering the MDA values at the end of the experiment, the results for the control group (1029.31 ± 183.83 g) were significantly lower than those of the DS group (1604.59 ± 412.06 g), while the levels in neither group differed significantly from the levels in the DB group (1172.85 ± 239.08 g).

HSP70 levels were measured for the third level of antioxidant defence. We did not find any differences between the groups during the evaluation of the HSP70 level (DS = 17.11 ± 1.26 g; DB = 17.05 ± 0.94 g; K = 17.03 ± 0.85 g), presumably because the fish did not experience heat shock. We speculate that the lack of difference in the production of heat shock proteins may also be due to the fact that the defence system had already coped with stress at the first two levels, and activation of the third level was not required.

## 4. Discussion

Several tank colours are recommended for European perch farming at different developmental stages. Although there are many studies about the different tank colours, the combination of these has not been investigated before. Therefore, the aim of this study was to evaluate the effects of combinations of different tank colours (grey; black bottom and grey sidewalls; grey bottom and black sidewall) on the production and antioxidant parameters of European perch. This study could assist European perch juveniles and market size fish producers in choosing the optimal tank colour combinations for rearing.

During the experiment, the survival rate was 100% for all experimental units. In our opinion, this favourable result could be due to the size of the fish and the optimal farming technology that was used [49].

Among the production parameters, significant differences were found in terms of the individual body weights and specific growth rates of the fish. The individual body weights of juveniles in the black-bottomed tanks were significantly higher than those in the tanks with black sidewalls and the control group. Compared with the DS group, significantly higher SGR values were found in the DB group. The individual body weights and specific growth rates of the fish, as well as cortisol levels, were strongly related. The cortisol levels were significantly lower in the DB group and the control group, compared with the DS group, which may indicate that the tank with the dark bottom did not trigger a stress response from the fish. In this study, the concentration of cortisol was not altered in the fish in the dark-bottomed tank, compared with those in the control group in the case of body weights. The increased cortisol levels signify a higher stress level in the fish organism [38]. In all cases, there was a close correlation between the amount of glucose in the blood and the cortisol levels of the fish. The fish in the DS group showed significantly higher glucose levels than those in the DB group. These values did not significantly differ from the control group. Some researches confirm our results, in which it was revealed that stress also affects fish growth, glucose, and cortisol levels—in both European perch and other fish species such as rainbow trout (*Oncorhynchus mykiss*), Yellow perch (*Perca flavescens*), and Common carp (*Cyprinus carpio*) [20,50,51]. Several studies have investigated the effects of different

tank colours on different fish species. De Abreu et al. [52] measured lower cortisol levels in zebrafish (*Danio rerio*) when kept in blue tanks, compared with white tanks.

In our study, we focussed on the contrast between two different colours (black and light grey), rather than the effects of the different colours themselves. Our results showed that of the different tested tanks, the highest stress was associated with the combination of dark walls with light-coloured bottoms.

The role of the catalase enzyme is to break down hydrogen peroxide molecules into water and oxygen. Hydrogen peroxide, which is a reactive oxygen species (ROS), generates a free radical molecule during oxidative stress. The higher the concentration of hydrogen peroxide, the more ROS are formed [32]. Significantly lower catalase levels were observed in the DB and control groups, compared with the DS group. This meant that the fish in the DS group experienced higher stress effects than those in the two other groups.

Among the vitamins, vitamin C is one of the most important antioxidant molecules [53,54]. Similar to the previously mentioned stress biomarkers, the concentration of vitamin C in the blood samples was found to be closely related to the individual fish body weights. The DB group had vitamin C levels that were more than three times higher than those of the DS group. Supporting our results, Narra et al. [55] stated that vitamin C has a beneficial effect on fish survival and growth. Lee et al. [56] achieved similar results when investigating the interactions and long-term effects of dietary vitamin C on the growth of yellow perch *(Perca flavescens)*. Based on their results, dietary vitamin C supplementation increases the growth rate. Based on our results, in the DB group, where the highest vitamin C concentration was measured, the fish were the least stressed. Presumably, they did not have to use vitamin C for glutathione metabolism.

Glutathione is one of the most important non-enzymatic antioxidants and is an important component of the defence system against oxidative stress. It is found mainly in its reduced form (GSH), although the oxidised form (GSSG) can also be observed. In the ascorbate–glutathione cycle, glutathione undergoes reduction by the enzyme glutathione reductase, which uses NADPH [57]. Glutathione is a cofactor of several enzymes involved in oxidative stress defence, including glutathione peroxidase (GPx) and glutathione reductase (GR) [32]. Higher GSSG and GR values ensure higher stress resistance. Our results showed that the DB and control groups had significantly higher GSSG and GR values than the DS group, which meant that the fish in these groups were less stressed. Together, GR and GPx comprise the GSH redox cycle [58]; therefore, a higher GR value implies a lower GPx level. Our measurements correspond to this statement.

MDA is a degradation product of lipid peroxidation. Its presence indicates the activity of free radicals in the body [59,60]. According to the results of the research of Tian et al., the production of MDA is influenced by light conditions [61]. Based on our results, fish kept in the light grey tank in the control group had the lowest MDA values, while the highest were found in the fish in the DS group. The levels in the DB group did not differ from either of the other groups. According to these results, a lighter tank colour promoted MDA production in the fish. In the groups with the highest levels of MDA, the animals had high levels of free radicals due to stress effects; therefore, lipid peroxidation was more intense in these groups.

Regarding the third level of the antioxidant system, we did not observe any difference in the production of HSP70 between the groups. It appeared as though the third level of the antioxidant system was not activated by the different stress factors during the experiment. Lucentini et al. [62] investigated the long-term effects caused by the expression of heat shock proteins in European perch. According to their results, temperatures above 20 °C may have long-term effects on European perch. In our experiment, the fish were kept at higher temperatures (22.65 ± 0.91 °C); nevertheless, we did not observe any differences related to this indicator.

## 5. Conclusions

Our study was focussed on the effect of the contrast between two colours (black and light grey) in different parts of the tank (bottom and sidewall) on the production

and antioxidant parameters of European perch during intensive rearing. The results of our experiment show that contrasts between the dark and light tank colours significantly influenced the production and antioxidant parameters of European perch. The tanks with black bottoms and light-grey sidewalls resulted in fish with significantly higher individual body weights but had no effect on the survival rate, feed conservation ratio, condition factor, or stock homogeneity. These results were also supported by the measurements of antioxidant parameters. Based on our results, the tanks with black bottoms and light-grey sidewalls could be favourable, because this combination of colours is probably similar to the natural habitat of European perch. In order to obtain more in-depth knowledge in this field, further studies are necessary.

**Author Contributions:** Conceptualisation, Á.M. and M.F.; methodology, Á.M., M.F. and A.K.; investigation, Á.M. and D.Z.H.; data curation, G.P.-A.; writing—original draft preparation, Á.M.; writing—review and editing, Á.M., M.F., J.R. and P.B.; supervision, M.F. and P.B.; project administration, M.F. and P.B.; funding acquisition, M.F. and P.B. All authors have read and agreed to the published version of the manuscript.

**Funding:** This research was funded by MAHOP-2.1.1-2016-2017-00002 (RESEARCHFISH). The APC was funded by EFOP-3.6.3-VEKOP-16-2017-00008 project.

**Institutional Review Board Statement:** The study was conducted in accordance with the Declaration of Helsinki, and approved by the Institutional Ethics Committee of University of Debrecen (DEMAB/15/2019).

**Acknowledgments:** This study was supported by the EFOP-3.6.3-VEKOP-16-2017-00008 project. The project was co-financed by the European Union and the European Social Fund. This research was supported by the MAHOP-2.1.1-2016-2017-00002 (RESEARCHFISH) project.

**Conflicts of Interest:** The authors declare no conflict of interest.

**Ethical Approval:** This study was approved and carried out in accordance with the local ethics committee's guidelines of the University of Debrecen under the registration number DEMAB/15/2019.

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
