# Peer review of "The Effects of Contrast between Dark- and Light-Coloured Tanks on the Growth Performance and Antioxidant Parameters of Juvenile European Perch (Perca fluviatilis)"

_water, doi:10.3390/w14060969_

Round 1

Reviewer 1 Report

In my opinion, this is a high-quality manuscript. Perfectly planned and conducted research. However, before accepting MS, it needs to be improved. My detailed comments are contained in the text of MS. To see them all, open the file in Acrobat Reader.

Main remarks:
1. when giving results in the text, units should be given.
2.The research in question concerns a fragment of the fattening - so this information should be provided both in the title and abstract, and in the aim of the study
3. Perch is indeed a very susceptible species to stress, but different developmental stages (eg larvae, juveniles, sub-adults) may have different environmental conditions, including different response to the color of swimming pool walls. The light gray walls and bottoms of the pools would never be recommended for larvae. Therefore, I think the paragraph about this should be presented either in the Introduction or in the Discussion.
4. This work is part of the modern research on the optimization of perch breeding and rearing under RAS conditions. A bit of information about this in the Introduction, and placing this research in this trend, would increase the value of MS and make it more complete.

I believe that the authors will improve this MS without any problems.

Reviewer 2 Report

The authors present the results of an experiment to understand the influence of light and colours (although it is grey vs black, so it is better to use dark- and light-coloured tanks like in the title) on the growth performance and antioxidant parameters of European perch.

While the experiment is generally well described, and the results are interesting, the paper is missing some rationale. It is evident from the abstract that rather than a theoretical research question, the authors have a descriptive aim. The theory behind their experimental design should be clearer. 

Below some line-by-line comments:

  • Lines 35-36. Remove one of the two currently.
  • Lines 37-39. From this sentence it is not clear how the 32,000 tonnes can be reached if the production is 954 tonnes. You need to rephrase the sentence. Maybe the 954 is the annual production and the 32000 is the total captures (in which time period)? 
  • Lines 42-44. What are optimal environmental conditions under intensive conditions? Need to clarify the sentence.
  • Lines 47-49. This sentence is too long and not clear, better to split into two sentences and clarify the concepts, with references.
  • Line 50. Add reference.
  • Line 51. You need to rephrase the sentences as "It has been shown that..." cannot be split.
  • Line 54. Why light and colour are so important? You are missing more context in your introduction. You are only making examples of some experiments but you need more theory.
  • Line 58-61. Here you introduce your experimental design, not your research question. You are missing the theoretical reasoning for your experiment. And you should really make predictions. Also, the aim of the study is usually at the end of the introduction not in the middle
  • Lines 62-80. Are these parts somehow related to your experiment? Not clear the link, you should make it clearer.
  • Lines 81-85. This should be part of your methods, this is not your research question. 
  • Line 101. Why light grey is considered as a control group rather than another experimental group?
  • Line 105. Is the measure of variability SE, SD, or CI?
  • Lines 123-125. Why did you use these parameters, they were not introduced (i.e., you need to present them in the Introduction). 
  • Lines 154-159. Levene's test is testing the homogeneity of the variances between groups not the homogeneity of data. And you should also specify that you run Levene's tests to check the assumptions. You did not specify, though, that you tested the assumption of normality that is the main assumption for ANOVA tests. Also, did you use Tukey or Duncan correction for post-hoc tests? It is confusing to see both of them in there. The data analysis section is not clear.
  • Tables. You should specify what is the difference between a and b.
  • In the results you need to specify if the variability is SD, SE, CI, or whatever.
  • Lines 218-219. Again this is the experimental design, the aim should be more theoretical. 
  • In the discussion, you should not repeat the results (especially not the numbers), you need to discuss your results. 
  • Line 326. The reference is missing the weblink and access date. 

Round 2

Reviewer 1 Report

I recommend accept the  MS after minor revision. I found only few places which should be corrected. My few comments are included in the text in the form of comments. To see them all, open the file in Acrobat Reader.

Reviewer 2 Report

The authors responded to all my comments. I do not have any further comments.